# The ACE2 Receptor from Common Vampire Bat (*Desmodus rotundus*) and Pallid Bat (*Antrozous pallidus*) Support Attachment and Limited Infection of SARS-CoV-2 Viruses in Cell Culture

**DOI:** 10.3390/v17040507

**Published:** 2025-03-31

**Authors:** Abhijeet Bakre, Ryan Sweeney, Edna Espinoza, David L. Suarez, Darrell R. Kapczynski

**Affiliations:** Exotic and Emerging Avian Viral Disease Research Unit, Southeast Poultry Research Laboratories, US National Poultry Research Center, United States Department of Agriculture, 934 College Station Road, Athens, GA 30605, USA; ryan.sweeney@usda.gov (R.S.); edna.espinoza@usda.gov (E.E.); david.suarez@usda.gov (D.L.S.); darrell.kapczynski@usda.gov (D.R.K.)

**Keywords:** SARS-CoV-2, ACE2, replication, model, bats

## Abstract

During the COVID-19 pandemic, severe acute respiratory syndrome coronavirus 2 (SC2) infection was confirmed in various animal species demonstrating a wide host range of the virus. Prior studies have shown that the ACE2 protein is the primary receptor used by the virus to gain cellular entry and begin the replication cycle. In previous studies, we demonstrated that human and various bat ACE2 proteins can be utilized by SC2 viruses for entry. Bats are a suspected natural host of SC2 because of genetic homology with other bat coronaviruses. In this work, we demonstrate that expression of ACE2 genes from the common vampire bat (CVB) (*Desmodus rotundus*) and the pallid bat (PB) (*Antrozous pallidus*), supports infection and replication of some SC2 viruses in cell culture. Two cell lines were produced, CVB-ACE2 and PB-ACE2, expressing ACE2 from these bat species along with human TMPRSS2, in a model previously established using a non-permissive chicken DF-1 cell line. Results demonstrate that the original Wuhan lineage (WA1) virus and the Delta variant were able to infect and replicate in either of the bat ACE2 cell lines. In contrast, the Lambda and Omicron variant viruses infected both cell lines, but viral titers did not increase following infection. Viral detection using immunofluorescence demonstrated abundant spike (S) protein staining for the WA1 and Delta variants but little signal for the Lambda and Omicron variants. These studies demonstrate that while ACE2 from CVB and PB can be utilized by SC2 viruses to gain entry for infection, later variants (Lambda and Omicron) replicate poorly in these cell lines. These observations suggest more efficient human adaption in later SC2 variants that become less fit for replication in other animal species.

## 1. Introduction

Severe acute respiratory syndrome coronavirus 2 (SC2), the causative agent of the COVID-19 pandemic, has evolved in humans over the past five years gaining increased transmissibility, with little or decreased change in virulence [1,2,3]. While SC2 has predominantly affected humans, other animal species have been shown to be naturally susceptible including such diverse species as white-tailed deer, cats, dogs, American mink, and tigers [4,5,6,7,8,9,10,11,12,13]. The true extent of susceptible animal species and their potential as reservoirs for SC2 remain unknown, but zoonotic spread from naturally infected animals has been documented [14,15,16]. Similarly, anthropozoonotic transmission of SC2 from infected humans to susceptible animals such as mink and zoo animals has also been suggested [14,17,18].

A key factor for the replication of SC2 in both humans and animals is the ability of the viral spike (S) protein to bind to angiotensin-converting enzyme 2 (ACE2) on the surface of target cells [19,20,21]. The ACE2 protein is found in most vertebrate species and the amino acid sequence is conserved among mammals. After receptor binding to ACE2, the viral S protein is cleaved by a host protease to allow the fusion of the viral envelope with the cell membrane to release the viral RNA into the cell. Although multiple host proteases appear to be capable of S protein cleavage, the host transmembrane protease serine 2 (TMPRSS2) appears to be the most important for this function [22,23]. Thus, ACE2 and TMPRSS2 sequence and expression in different tissues are the primary determinants of SC2 host range in animals.

Bats represent an interesting reservoir and vehicle for disease transmission. With approximately 1400 species, some of which can live up to 40 years, bats account for approximately 20% of all mammalian species [24,25,26,27]. As the only mammals capable of self-powered flight, bats have evolved to thrive in unique ecological niches, consume varied food sources, and are nearly ubiquitous across the planet [24]. Owing to their unique adaptations, bats exhibit a diverse phylogeny.

Previous studies have demonstrated that bats are a reservoir for a number of important viruses [28]. Bats have been shown to harbor severe acute respiratory syndrome coronavirus-1 (SARS-CoV-1) and Middle East respiratory syndrome (MERS) virus, and they have been speculated to be the original reservoir for SC2 [29,30,31,32,33]. Fruit bats of the genus *Pteropus* are reservoirs of both Nipah and Hendra viruses, and they can transmit these to intermediary hosts such as swine and horses [34]. Fruit bats of the genus *Rousettus* can harbor Nelson Bay reovirus and Marburg virus, and serological evidence suggests that other bat species may support infection by the Ebola virus [35,36,37]. Additionally, of 4800 coronaviruses detected, more than thirty percent were of bat origin [38]. However, only 543 of the ~1400 bat species have been sampled for coronaviruses, suggesting that the true diversity of bat coronaviruses is likely to be more extensive than currently understood [39]. Interestingly, most bat species survive viral infection and present no symptoms of disease, indicative of a unique immune system that allows bats to tolerate infectious diseases that are lethal to other mammalian species [24].

The common vampire bat (CVB), prevalent from South America up to Northern Mexico in North America, can carry rabies and dengue and is associated with the transmission of both viruses to humans in Brazil and Mexico [40,41,42,43]. Because the CVB, a strictly sanguivorous bat, feeds on large animals such as white-tailed deer, which are naturally infected with SC2, it is possible that CVBs could be exposed to and transmit SC2 among animals during feeding [44,45,46,47,48,49,50,51]. Further, environmental modeling predicts that cattle in Mexico, Central America, Venezuela, and Colombia are at a higher risk of CVB-transmitted disease outbreaks [52]. Pallid bat (PB) habitats are widespread ranging from central Mexico to British Columbia [53]. They have also been demonstrated to be a reservoir for skunk rabies virus variant [39,54,55]. While PBs primarily feed on insects and scorpions, they can also feed on wild rodents such as deer mice which have been demonstrated to be susceptible to SC2 [56]. One study determined that despite geographical clustering, PB migration could increase the potential for disease movement and transmission [53].

Despite the importance of bats in the disease ecology of some pathogens, in vivo bat models to study disease infection and transmission are uncommon because working with bats is resource-intensive. To overcome this challenge, we propose to use a cell culture model that can recapitulate the susceptibility to SC2 infection in different host species. Kapczynski et al. previously established an experimental model where non-permissive chicken DF-1 fibroblast cells were engineered to constitutively express ACE2 and TMPRSS2 genes from different species [57]. Both human ACE2 and TMPRSS2 expression was essential to support SC2 infection and replication. Further, cell lines expressing ACE2 and TMPRSS2 from seven additional animal species (cat, horse, pig, goat, golden hamster, little brown bat, and great roundleaf bat) were constructed of which only cat, golden hamster, and goat supported SC2 replication. Briggs et al. recently extended these studies to individually express ACE2 genes from different bat species using a DF1 cell line that also expressed human TMPRSS2 protease [58]. Results demonstrated variable ability of these different bat ACE2 genes to support SC2 infection and replication in cell culture. Interestingly, the original WA1 lineage virus replicated to higher titers compared to other variants tested. In this study, we further used this model to examine the ACE2 gene from two additional bat species to determine if SC2 viruses can use these proteins for attachment and infection. These studies further support application of an in vitro model using engineered chicken DF-1 fibroblast cells to test susceptibility of additional bat species to SC2.

## 2. Materials and Methods

### 2.1. Viruses Used and Growth 

The following coronaviruses were used in this study and were obtained from the BEI research resources repository, the National Institute of Allergy and Infectious Diseases (NIAID), and the National Institutes of Health (NIH): Washington strain (WA1) (USA/WA1/2020/Wuhan lineage (BEI NR-52286)), Delta strain (USA/PHC658/2021/B.1.617.2 (BEI NR-55611)), Lambda strain (Peru/un-CDC-2-4069945/2021/Lineage C.37 (BEI NR-55654)) and Omicron (hCoV19/USA/mD-HP20874/2021 (BEI NR-56461)) [59]. All infectious work was carried out in a biosafety level-3 enhanced laboratory following protocols approved by the U.S. National Poultry Research Center Institutional Biosafety Committee. Viral stocks were obtained from BEI Resources and then scaled up into mycoplasma-free Vero E6 cells as per standard protocols [57]. DF-1 cells expressing human ACE2 and human TMPRRS2 that support productive replication of all the four coronavirus variants studied here were used as a positive control.

### 2.2. Cell Culture

DF-1 (chicken fibroblasts) (ATCC-UMNSAH-CRL-3586) and Vero E6 (ATC-CRL-1586) cells were grown in Dulbecco’s modified Eagle’s Medium (DMEM) supplemented with 1 mM Sodium pyruvate, high glucose, 1X Antibiotic-antimycotic mix (Corning, AZ, USA), and 10% heat-inactivated fetal bovine serum (BioWest, Lakewood Ranch, FL, USA). Cells were periodically tested to ensure they were mycoplasma-free. Both cell lines were grown under 95% humidity and 5% CO_2_; DF-1 cells were grown at 39 °C while Vero E6 cells were grown at 37 °C.

### 2.3. Bat ACE2 Plasmid Construction Using the PiggyBac Transposon Vector

ACE2 sequence from CVB (XM_024569930.3) (CVB-ACE2) and PB (accession number MT333480.1) (PB-ACE2) were de novo synthesized commercially into PiggyBac^®^ transposon expression plasmids (VectorBuilder, Chicago, IL, USA) used to obtain construct VB220322-1335atd and VB220322.1-1341jmh, respectively. Each of these constructs expressed the corresponding bat ACE2 gene under a constitutive immediate early CMV promoter along with enhanced green fluorescent protein (EGFP) [60,61]. Glycerol stocks of constructs in *E. coli* were obtained from VectorBuilder (USA) and streaked on Luria–Bertani agar plates (BD Biosciences, San Jose, CA, USA) containing 100 µg/mL of sterile Ampicillin prepared in house (ThermoScientific, Suwanee, GA, USA). Single colonies were lifted and inoculated into Luria–Bertani broth (BD Biosciences, San Jose, CA, USA), incubated overnight, and cultures were used for plasmid preparation using Monarch plasmid miniprep kits (New England Biolabs (NEB), Ipswitch, MA, USA) as per manufacturer’s instructions. Plasmids were digested with NcoI enzyme (NEB, USA) to validate presence of inserts. After overnight primary culture, verified clones were stored as 25% glycerol stocks.

### 2.4. Transfection of Bat ACE2 Plasmids in E. coli

For transfection of bat ACE2 plasmids above, glycerol stocks were plated on fresh Luria–Bertani (BD Biosciences, USA) agar plates as above, and single colonies were inoculated into 5 mL of Terrific broth (BD Biosciences, USA) containing 50 µg/mL of sterile Ampicillin prepared in house (ThermoScientific, USA) and grown for 16 h with shaking at 225 rpm at 30 °C in an incubator shaker (Amerex, Concord, CA, USA). Primary cultures at 1% inoculum volume were added into 100 mL of Terrific broth containing 50 µg/mL of sterile Ampicillin, and cultures were shaken overnight at 225 rpm and 30 °C. Cells were pelleted at 6500× *g* and then used for isolating endotoxin-free plasmid DNA using a Nucleobond Extra Maxi EF kit (Machery-Nagel, Allentown, PA, USA) as per the manufacturer’s instructions. Plasmid DNA was quantified using a Nanodrop 2000 (ThermoFisher Scientific, Suwanee, GA, USA).

### 2.5. Transgenic Cell Line Development Using the PiggyBac Plasmid Transposon System

DF1 cells expressing human TMPRSS2 (hTMPRSS2) and RFP under a constitutive CMV promoter were previously generated using a lentiviral transduction system [57]. These cells were transfected with hyperactive PiggyBac transposase (HyBase) and CVB-ACE2 or PB-ACE2 transposon plasmids in a 1:5 ratio with Xfect transfection reagent (Takara-Bio, San Jose, CA, USA) as per the manufacturer’s recommended protocol. Complete media from monolayers in 6-well plates were replaced with DMEM and the transfection complexes for 4 to 16 hrs. Transfection complexes were replaced with fresh complete media containing 10% FBS and incubated for 48–72 h to validate EGFP and RFP expression using fluorescence microscopy (EVOS 5000, ThermoFisher Scientific, USA).

### 2.6. Fluorescence-Activated Cell Sorting (FACS)

Transfected cells from 6-well plates were scaled to T25 and T75 flasks (90% confluency) using routine subculture and validating EGFP and RFP expression at each passage. Cells were trypsinized using 0.025% Trypsin-EDTA, filtered through a 50 µm cell strainer (ThermoFisher Scientific, USA) and then sorted using a Bio-Rad S3e cell sorter (BioRad, Hercules, CA, USA) at the flow cytometry core at the University of Georgia. GFP and mCherry red double-positive cells were enriched three to four times by flow sorting to obtain a population of DF-1 cells expressing hTMPRSS2 and CVB-ACE2 or hTMPRSS2 and PB-ACE2 (Appendix A).

### 2.7. RNA Extraction and qRT-PCR Conditions for Confirmation of Bat ACE2:hTMPRSS2 Cell Lines

Total RNA from wild-type DF-1 cells or CVB-ACE2 and PB-ACE2 cell lines was isolated using RNAzol RT (MRCgene, Cincinnati, OH, USA) following the manufacturer’s protocol. Total RNA was quantified using either Nanodrop 2000 (ThermoFisher Scientific, USA) or Qubit BR RNA assays (Invitrogen, ThermoFisher Scientific, Suwanee, GA, USA) as per the manufacturer’s recommendations. Equal amounts of total RNA were used in first-strand cDNA synthesis using Maxima H minus RT (ThermoFisher Scientific, USA) as per the manufacturer’s recommendations. ACE2 and hTMPRSS2 expression in these samples was determined using human ACE2, bat ACE2, and chicken GAPDH primers as previously published [57]. ACE2 expression was determined by using primers ACE2 Universal short F-5′-GCC, AAG, GAA TTT TTG GAC AAG TTT AAC-3′ AND ACE2Uni-shortR 5′-TGG AAT TTG AGA TGT CAC ATT TG-3′.

### 2.8. ACE2 and TMPRSS2 Protein Analysis by Western Blotting

Protein expression of *Dr*-ACE2, *Ap*-ACE2 and hTMPRSS2 was confirmed by Western blot analysis. Briefly, lysates from 10^6^
*Dr*-ACE2 or *Ap*-ACE2 cells were prepared using RIPA Lysis buffer (Catalog # R0278, Millipore-Sigma, Burlington, MA, USA) as per manufacturer recommendations. Protein was quantified using BCA assay (Catalog # 23225, Pierce, ThermoFisher Scientific, USA). Ten micrograms of total protein estimated was boiled with 2X Laemelli buffer with b-mercaptoethanol at 95 °C for 5 min, cooled on ice and then loaded on a 4–20% Mini-PROTEAN TGX Precast gel (Catalog # 4561093, BioRad, Hercules, CA, USA) along with protein standards (Catalog # P7719S, NEB, USA). Samples were resolved in 1X Tris-Glycine SDS buffer (Catalog # 1610732, BioRad, Hercules, CA, USA) for 25 min at 100V constant voltage. Bands were transferred to Immunoblot PVDF membrane (Catalog # 1620177, BioRad, Hercules, CA, USA) in home-made Towbin buffer at 15V for 25 min using Trans-Blot SD Semi-Dry transfer cell (Catalog # 1703940, BioRad, Hercules, CA, USA). Blots were blocked overnight in 1X TBST with 3% BSA at 4 °C with gentle agitation. Blots were consequently incubated with respective primary monoclonal antibodies in TBST with 3% BSA either for 1 h at room temperature or 4 °C overnight. Primary antibodies included mouse anti-human ACE2 (1:1500 dilution) (Catalog # TA803844, Origene, Rockville, MD, USA), rabbit anti-human TMPRSS2 (1:1000), (Catalog # ab10913, AbCam, Cambridge, UK) and mouse anti-beta actin (1:2000) (Invitrogen, San Jose, CA, USA). Blots were washed 3X TBST for 5 min each wash. Washed blots were then incubated with goat anti-mouse IgG-coupled to horseradish peroxidase (HRP) (1:5000) (Catalog # sc-2005, Santa Cruz Biotechnology, Dallas, TX, USA) or goat anti-rabbit IgG-HRP (1:5000) (Catalog # sc-2030, Santa Cruz Biotechnology, Dallas, TX, USA) for 1 h at 37 °C with gentle agitation. Blots were finally incubated with Supersignal west Femto maximum sensitivity substrate (Catalog # 34095, Pierce ThermoScientific, Waltham, MA USA) for 5–10 min as per manufacturer’s suggestions and then imaged using a G:Box mini6 (Syngene International Ltd., Bengaluru, India).

### 2.9. Immunofluorescence Analysis of Viral Infection

CVB-ACE2, PB-ACE2, or control DF-1 cells were seeded overnight in I-bidi 8-well chamber slides (ThermoFisher, Carlsbad, CA, USA), allowed to become 70% confluent, and then infected with respective viruses for 48 h at an MOI of 1.0. After 48 h, media was removed, and cells were fixed using 1:1 ice-cold ethanol: methanol mixture for 5 min. Slides were washed thrice with ice-cold PBS, then blocked in blocking buffer for 1 h at room temperature, and then washed again with PBS thrice at room temperature. Cells were next incubated with rabbit anti-S mAb (Origene, Rockland, MD, USA) at 1:250 dilution. Cells were washed thrice with PBS and then incubated with Cy3 conjugated goat anti-rabbit IgG H + L (Abcam, Cambridge, UK) at 1:500 dilution in PBS for 1 h. Secondary antibody was removed by aspiration, and slides were washed thrice in ice-cold PBS. Nuclei were counterstained with 4′,6-diamidino-2-phenylindole (DAPI) (1 µg/mL) in PBS [62]. Images were captured on an EVOS 5000 microscope using default settings at 10X magnification. Raw image (8-bit or 16-bit RGB) files were analyzed using ImageJ (version 1.54f) [63]. The percentage of S-positive cells in the images above was calculated using Image J by splitting each color image into individual color channels, adjusting thresholds, and processing for segmentation and particle count analysis.

### 2.10. Infection and Replication Dynamics of SC2 and Variants in CVB-ACE2, PB-ACE2 Cell Lines

CVB-ACE2 or PB-ACE2 dual sorted cells were plated in 6-well plates overnight (5 × 10^5^ cells per well). Cells were washed with PBS twice before infection with SC2 of WA1, Delta, Lambda, and Omicron variants at an MOI of 1.0 in triplicate wells. Viruses were allowed to adsorb for 1 h, inoculum was then replaced with complete media, and virus growth was monitored for 72 h. Supernatants (200 µL per well) were collected at 0, 6, 24, 48 and 72 h post inoculation (h.p.i.) for detection of replicating virus by qRT-PCR [57]. Cytopathic effect (CPE) was monitored by microscopy using an EVOS 5000 microscope (ThermoScientific, Suwanee, GA, USA).

### 2.11. ACE2 Sequence Analysis

Nucleotide sequences of ACE2 genes were downloaded from the National Center for Biotechnology Information (NCBI) (https://www.ncbi.nlm.nih.gov/nucleotide accessed on 28 March 2022) nucleotide database. Accession numbers are provided in Appendix A. Multiple sequence alignments of CVB and PB ACE2 with previously studied mammalian, bat, and chicken ACE2 genes were performed using the Jukes–Cantor genetic distance model with 500 bootstraps, and phylogenetic trees were built using neighbor-joining algorithm in Geneious Prime (version 2024.0.4).

### 2.12. Statistics

Statistical significance between viral titers was performed at 48 h.p.i. as previously described with 2-way ANOVA using Tukey’s multiple comparison test in GraphPad Prism (ver. 10.2.1) [57]. All comparisons included three or more replicates of samples compared and post hoc analyses.

## 3. Results

### 3.1. Common Vampire Bat and Pallid Bat ACE2 Sequences Cluster Independently

Phylogenetic distances between these ACE2 sequences were computed using the Jukes–Cantor model with 500 bootstraps followed by tree building with a neighbor-joining algorithm. These analyses demonstrated that most bat sequences clustered together as expected of sequences with a common ancestry (Figure 1). The conservation plot identified three main regions in bat ACE2 proteins that were 100% identical across sequences (Appendix A).

These included regions encompassing amino acid (aa) 265–273, 376–380 and 401–407 (Appendix A).

We aligned full-length ACE2 amino acid sequences of CVB, PB, mammalian (human, pig, goat, and horse), and previously studied bats using Clustal W [57,58,64,65]. The mean length of most bat ACE2 sequences was 806 aa. Inter-species comparisons of ACE2 proteins from CVB and PB were conducted, and ACE2 demonstrated a mean identity of 78.2% and 79.6%, respectively, with other sequences studied (Appendix A). Chicken ACE2 was approximately 64.5% identical to other ACE2s studied. CVB and PB ACE2s demonstrated 79.5% and 79.2% identity with human ACE2 (Appendix A) but only 63.8 and 62.9% identity with chicken ACE2 [58].

### 3.2. Cells Lines Expressing CVB or PB ACE2 Support Replication of Multiple SC2 Variants

Previously, we performed control experiments with just DF-1 cells that lack ACE2 or TMPRSS2 and observed no replication of SC2 variants in wild-type DF-1 cells [57]. In this study, DF-1 cell lines expressing either the CVB-ACE2 (Appendix A) or PB-ACE2 (Appendix A) genes were established and purified using flow cytometry with expression of ACE2 confirmed by qRT-PCR (Appendix A) and Western blot (Appendix A). In prior studies, peak titers were observed at 48 h post infection; hence, this time point was chosen to compare replication across variants and cell lines [57,58].

In CVB-ACE2 cells, infection with both WA1 and Delta variants of SC2 peaked to approximately 10^4^ TCID_50_/mL at 48 h (Figure 2A), and titers were significantly lower with the Omicron viral variant. Lambda variant peaked to ~10^3.5^ TCID_50_/mL at 24 h post infection and remained at these levels through later time points. Differences in viral titer between WA1 or Delta versus Lambda variants at 48 h post infection were not significant. Lambda variant titers were also not significantly different relative to Omicron variant titers. Omicron variant titers remained lower at ~10^3^ TCID_50_/mL, a log fold lower compared with WA1 and Delta (Figure 2A and Appendix A).

In PB-ACE2 cells, WA1 and Delta virus variant titers peaked to ~10^5^ TCID_50_/_mL_ at 48 h post infection and then reduced significantly by 72 h post infection (Figure 2B). Lambda virus titers peaked to 10^4^ TCID_50_/_mL_ by 24 h but did not increase further at 48 h and 72 h post infection. Titers for the Omicron lineage virus did not increase beyond ~10^3^ TCID_50_/_mL_ at any time point, suggesting that the Omicron variant was unable to successfully replicate in PB-ACE2 cells (Figure 2B and S3B). Between CVB-ACE2 and PB-ACE2 cells, PB-ACE2 better supported the replication of WA1 and Delta SC2 variants.

In general, viral titers observed in cells expressing bat ACE2 were lower when compared to those expressing human ACE2 (Figure 2C and Appendix A). However, differences in insertion systems with the bat ACE2 genes compared to the human ACE2 gene may explain these differences. The bat system utilized transposon insertion while the human used lentivirus insertion, which may affect protein expression [57,58].

Viral infection in both CVB-ACE2 (Figure 3A–D) and PB-ACE2 (Figure 4A–D) cells was confirmed by immunofluorescence staining with anti-S antibody. Fluorescent microscopy demonstrated a strong anti-S signal in CVB-ACE2 expressing cells infected with WA1 (Figure 3A) and Delta (Figure 3B) virus variants. Signal intensity was reduced in Lambda virus-infected cells (Figure 3C). Little S staining was observed for Omicron-infected CVB-ACE2 cells (Figure 3D). Similarly, WA1 (Figure 4A) and Delta (Figure 4B) virus variants infected PB-ACE2 cells and demonstrated strong S staining, while the reduced fluorescent signal was observed in both Lambda (Figure 4C) and Omicron (Figure 4D) variant-infected cells. 

Taken together, these data demonstrate that CVB and PB ACE2 support infection of two of the SC2 viruses used here. However, the more recent Lambda and Omicron variants did not replicate as determined by TCID_50_, number of infected cells, and virus staining.

## 4. Discussion

Since the beginning of the SC2 outbreak, animal infection studies have been performed to establish models of human infection and to determine the natural host range of the virus. However, animal challenge studies are difficult to perform because of the requirement for BSL-3 containment facilities and it is impractical to test every possible susceptible species. Alternative approaches to predict susceptible species have included bioinformatic analyses and cell culture model systems [66,67,68,69,70]. Kapczynski et al. established an in vitro DF-1 model in which cells were genetically engineered to express ACE2 and TMPRSS2 genes from various animal species to test for attachment and infection by SC2 viruses [57]. This model identified several animal ACE2 genes that supported virus binding and replication suggesting that these species could serve as reservoirs and/or directly transmit the virus. A positive correlation has been observed between cell culture data and natural or experimental infections of the species tested [57,66,71]. This cell culture model system is a useful method to screen animal species’ susceptibility to infection and inform future epidemiologic and experimental challenge studies to potentially susceptible species.

Because CVB and PB are both prevalent in North America, we tested whether their ACE2 genes could support the attachment and replication of SC2 viruses. This manuscript builds on the previous report by Briggs et al., where the authors examined SC2 infection using ACE2 from seven different bat species [58]. We demonstrate here that the CVB and PB ACE2 sequences support the binding of the original WA1 and Delta variants. While the Lambda and Omicron lineage viruses were able to attach and infect both cell lines, they were unable to sustain a productive infection. This is consistent with previous findings [58].

In these studies, human TMPRSS2 was expressed in all cell lines to eliminate it as a variable of the study in constructing the bat ACE2 cell lines. This also overcame the issue that many bat genomes are not well annotated such that nucleotide sequences for bat TMPRSS2 may not be available. Further, differences between bat and chicken ACE2 were previously observed and may explain why chickens remain resistant to SC2. Consistent with the viral titers, we also observed higher S protein staining among CVB-ACE2 and PB-ACE2 cells for WA1 and Delta compared to Lambda or Omicron variant infected cells. Poor S staining with the Omicron variant corroborates the poor or lack of replication of Omicron variant viruses in CVB-ACE2 and PB-ACE2 cells, respectively. Previous studies demonstrated that Omicron variants of SC2 have accrued mutations in the non-structural protein 6 (nsp6), which is believed to reduce replication fitness and kinetics [72]. Omicron variant viruses were recently demonstrated to be significantly attenuated in a feline model, which may lend additional support for this hypothesis [71]. Finally, although the Omicron S protein has a higher binding affinity for human ACE2, Omicron virus variants have been demonstrated to use TMPRSS2 less efficiently leading to lower replication and virulence [1,73,74].

Other factors that can contribute to differential infectivity include inefficiency of S protein cleavage and poor fusion of the viral envelope with the host cell membrane [75]. In studies in human cells, the S proteins of eleven recent SC2 variants, including Delta and Lambda, demonstrated increased fusion with host cells and higher titer replication. In contrast, Omicron S protein demonstrated less cleavage and fusion compared to other variants. Finally, the study demonstrated a strong correlation between fusogenicity, S1/S2 cleavage and plaque size as an indicator of viral fitness [75]. Thus, increased fuso-genicity was positively correlated with better infectivity and consequently better replication in this in vitro model.

In conclusion, these studies demonstrate that ACE2 genes from both the CVB and PB can support infection and replication of some SC2 variants, though to varying extent. The infectivity and replication kinetics of WA1 and Delta were comparable, while Lambda and Omicron did not replicate as well. WA1 and Delta virus variants may thus be able to maintain a niche in CVBs and PBs; however, it also suggests that more recent variants of the Lambda and Omicron lineages may not be able to sustain fitness in CVB and PB populations. These studies further demonstrate the usefulness of the in vitro model for testing the host range of emerging variant viruses during pandemics and outbreaks.

## Figures and Tables

**Figure 1 viruses-17-00507-f001:**
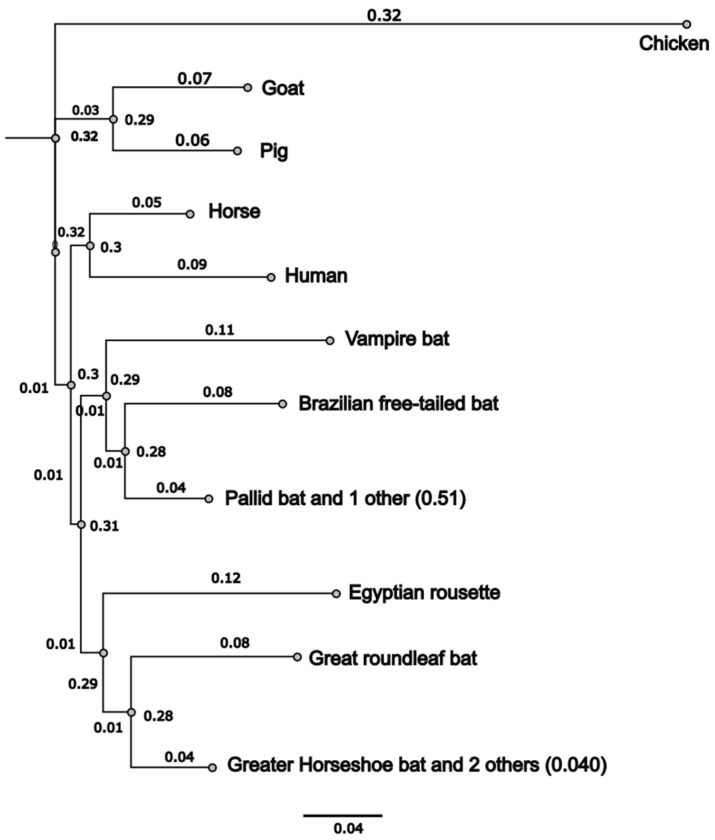
Phylogenetic tree of bat ACE2 protein sequences with mammalian or avian ACE2 sequences. Phylogenetic trees were constructed by aligning CVB and PB ACE2 protein sequence with previously studied mammalian or bat ACE2 sequences. Alignments were performed using the Jukes–Cantor genetic distance model, tree was built using neighbor-joining algorithm using Chicken-ACE2 as an outgroup since it does not bind to SC2 S protein. Consensus tree was generated by resampling with 500 bootstraps. Branch labels indicate substitutions per site. Scale bar is shown.

**Figure 2 viruses-17-00507-f002:**
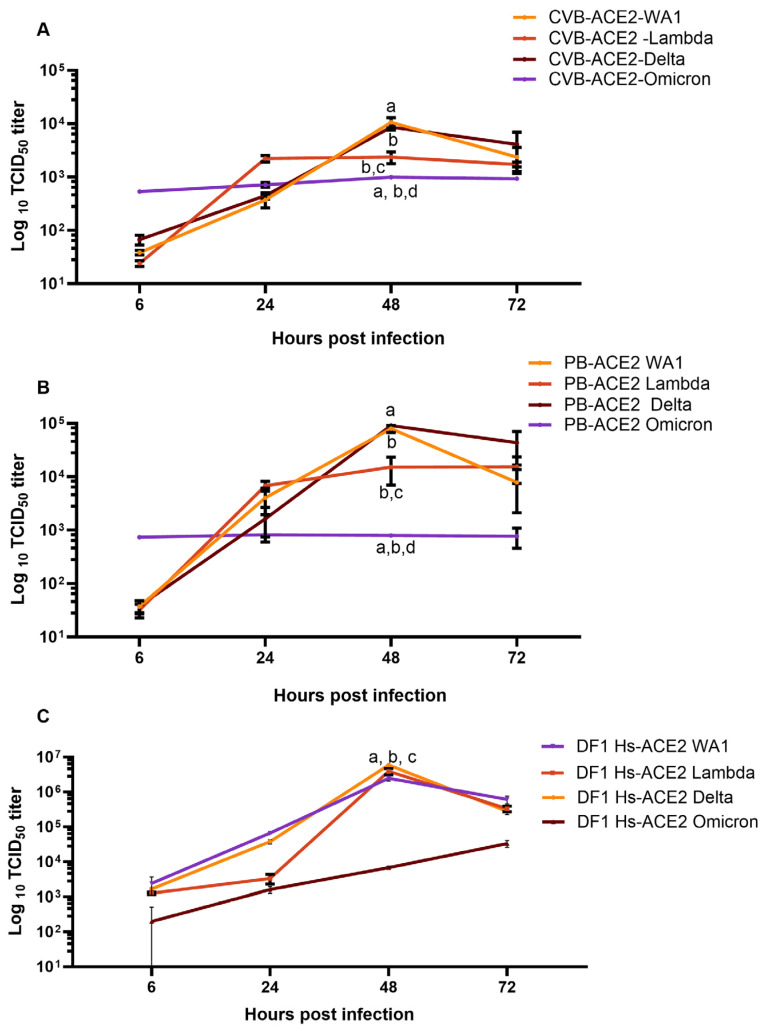
CVB-ACE2 and PB-ACE2 cells support infection of SC2 variants. Line graphs show log_10_TCID_50_ titers/mL of WA1, Delta, Lambda, and Omicron lineage SC2 viruses in CVB-ACE2/hTMPRSS2 (**A**), PB-ACE2/hTMPRSS2 expressing DF-1 cells (**B**) and DF-1 cells expressing human ACE2 and TMPRSS2 (**C**). Data represent mean ± SD from three independent experiments for each time point. Statistical comparisons were conducted with 2-way ANOVA using repeated measures with Geisser–Greenhouse correction and Tukey multiple comparisons test with individual variances computed for each comparison. Lines with different lowercase letters indicate statistically significant differences (*p* < 0.05). Titers are indicated on Y-axis and time points of infection are indicated on the X-axis. CVB = Common vampire bat *(Desmodus rotundus*), PB = Pallid bat (*Antrozous pallidus*), Hs = *Homo sapiens*.

**Figure 3 viruses-17-00507-f003:**
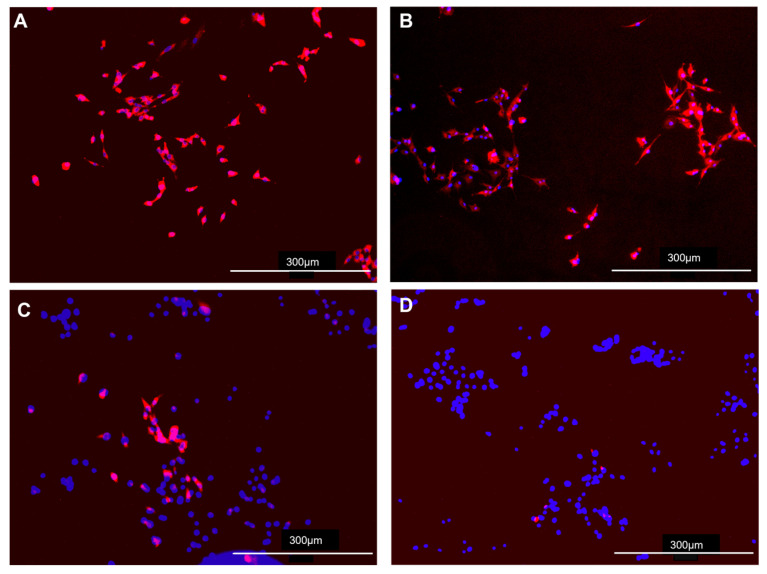
Immunofluorescence microscopy of SC2 infected CVB-ACE2 cells. Confluent (75%) monolayers of CVB-ACE2 expressing DF-1 cells on iBID chamber slides were infected with WA1 (**A**), Delta (**B**), Lambda (**C**) or Omicron (**D**) variant of SC2 for 48 h and then stained for SC2 S protein and counterstained for nuclei using DAPI as stated in materials and methods. Scale bars at the bottom right represent 10× magnification.

**Figure 4 viruses-17-00507-f004:**
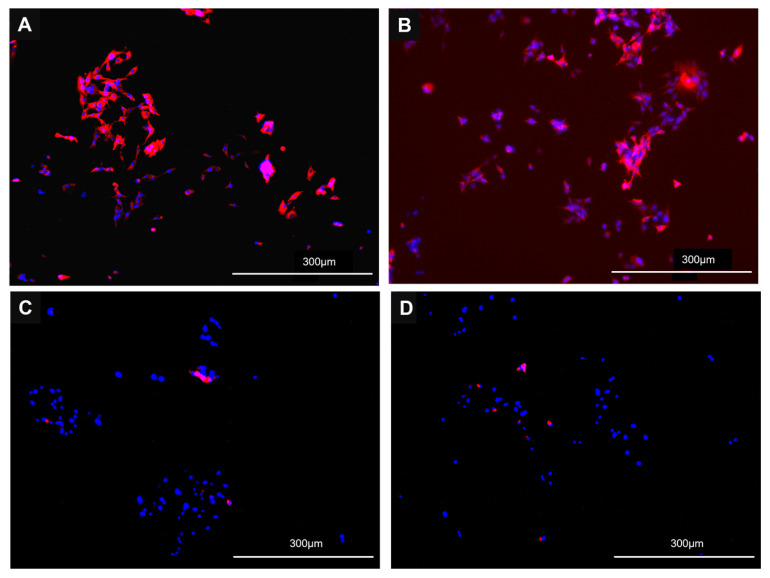
Immunofluorescence microscopy of SC2 infected PB-ACE2 cells. Confluent (75%) monolayers of PB-ACE2 expressing DF-1 cells on iBID chamber slides were infected with WA1 (**A**), Delta (**B**), Lambda (**C**) or Omicron (**D**) variant of SC2 for 48 h and then stained for SC2 S protein and counterstained for nuclei using DAPI as stated in materials and methods. Scale bars at the bottom right represent 10× magnification.

## Data Availability

All data associated with this manuscript will be deposited into AgData commons (https://www.nal.usda.gov/services/agdatacommons).

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
