# Peer review of "The ACE2 Receptor from Common Vampire Bat (Desmodus rotundus) and Pallid Bat (Antrozous pallidus) Support Attachment and Limited Infection of SARS-CoV-2 Viruses in Cell Culture"

_viruses, 2025, doi:10.3390/v17040507_

Round 1

Reviewer 1 Report

Comments and Suggestions for Authors

In this short manuscript, Bakre. et al. generate chicken fibroblast cell lines (DF-1) with exogenous expression of bat ACE2 receptor from two bat species (common vampire bat and pallid bat) to evaluate their influence of bat ACE2 in SARS-CoV-2 infectivity. This study is similar to a previous study previously published by the authors (PMID: 37542819). However, here authors evaluate ACE2 influence on infection from two bat species that were not tested before. Below you can find some issues that I have detected after reading the manuscript:    

MAJOR

  • Figure 2 must be improved, as the main findings of this study are shown here. First, a negative control for infection is needed. This control should be wild-type or mock-transfected DF-1 cells infected (as done previously in PMID: 37542819) with the different SARS-CoV-2 variants to compare the viral titers obtained in cells without bat / human ACE2 expression vs. cells expressing bat ACE2 expression. Second, the data could be represented in a different way to show the impact of bat ACE2 expression on infection. This would be achieved by representing  the viral titer obtained in each cell line (DF-1, DF-hACE2, CVB-ACE2, PB-ACE2) at the different timepoints within the same plot (including one plot per variant). The current way of representation allows for a better comparison of the infectivity of each variant for the same cell line, but makes it difficult to visualize the impact of bat ACE2 on infection.
  • I have several concerns about the detection of ACE2 expression via western-blotting. The information about how western blotting was performed is missing in Materials and Methods. Similarly, there is not a reference about which anti-ACE2 antibody was used to detect expression of bat ACE2. According to the results shown in Supp. Fig. 2C, I consider that this antibody might not be working properly, since the band intensity is very low. Is this antibody specific for bat ACE2? If not, how conserved is the epitope in bats when compared to the target species of this antibody?
  • Authors explain in line 242 that ACE2 expression is evaluated via RT-qPCR but they do not include a figure summarizing this result. This information would be useful considering how weak ACE2 detection is via western-blotting.
  • I consider that the conclusion obtained from this study are correct according to the obtained results, however it seems contradictory with their own previous study (“SARS-CoV-2 utilization of ACE2 from different bat species allows for virus entry and replication in vitro”, PMID: 37542819).

MINOR

  • Supplementary Figure 1, please indicate within the figure the three main regions with 100% shared identity that are mentioned in lines 225-226 using a red box or other visual element.
  • Lines 228-229: is the extent of the regions described correctly? For instance, first region seems to extend from position 266 to 277 according to the identity color code.
  • Supplementary Figure 2 A and B are a composite of fluorescence acquired from two different channels in red (human TMPRSS2) and green (bat ACE2). Please indicate within each picture frame what are the colors represents to simplify the understanding of the pictures. Also, the scale bar value is too small.
  • A MOI = 1 is an abnormally high MOI to use for SARS-CoV-2 infection. Why authors used such a high MOI?
  • Lines 261-264, authors explain that differential infectivity between human and bat ACE2 cell lines can be explained by the use of different insertion systems, which is true. However, the affinity of bat ACE2 for SARS-CoV-2 RBD is significantly lower than human ACE (reported in PMID: 33335073). This is another parameter that should be taken into account, which in my opinion, has more influence than the transfection system.
  • Include in Materials and Methods which primer sequences were used to evaluate ACE2 expression.
  • Lines 282-284, the text located here must be removed from the final version.

Author Response

In this short manuscript, Bakre. et al. generate chicken fibroblast cell lines (DF-1) with exogenous expression of bat ACE2 receptor from two bat species (common vampire bat and pallid bat) to evaluate their influence of bat ACE2 in SARS-CoV-2 infectivity. This study is similar to a previous study previously published by the authors (PMID: 37542819). However, here authors evaluate ACE2 influence on infection from two bat species that were not tested before. Below you can find some issues that I have detected after reading the manuscript:    

We thank the reviewers for their time and effort in improving the manuscript. Our responses to the comments are listed in blue below.

Reviewer 1.

MAJOR

  • Figure 2 must be improved, as the main findings of this study are shown here. First, a negative control for infection is needed. This control should be wild-type or mock-transfected DF-1 cells infected (as done previously in PMID: 37542819) with the different SARS-CoV-2 variants to compare the viral titers obtained in cells without bat / human ACE2 expression vs. cells expressing bat ACE2 expression.

We performed control experiments with just DF-1 cells that lack ACE2 or  TMPRSS2 and observed no replication of SC2 variants in wild type DF-1 cells as shown previously.  In contrast, robust  infection and replication of SC2 variants was observed in DF-1 cells expressing hACE2 and hTMPRSS2 (Figure 2C).

  • Second, the data could be represented in a different way to show the impact of bat ACE2 expression on infection. This would be achieved by representing  the viral titer obtained in each cell line (DF-1, DF-hACE2, CVB-ACE2, PB-ACE2) at the different timepoints within the same plot (including one plot per variant). The current way of representation allows for a better comparison of the infectivity of each variant for the same cell line, but makes it difficult to visualize the impact of bat ACE2 on infection.
  • We thank the reviewers for their time and effort in improving the manuscript. We have plotted the data as requested as well and show in Supplementary Figure 3, the viral titers across time points for each SC2 variant virus in each cell line. The data clearly show that virus titers peak at 48h post infection in both cell lines for the WA1 and Delta SC2 variant and 24-48h for the lambda SC2 variant. Omicron titers either only increased very slowly at these time points or remained static.

  • I have several concerns about the detection of ACE2 expression via western-blotting. The information about how western blotting was performed is missing in Materials and Methods. Similarly, there is not a reference about which anti-ACE2 antibody was used to detect expression of bat ACE2. According to the results shown in Supp. Fig. 2C, I consider that this antibody might not be working properly, since the band intensity is very low. Is this antibody specific for bat ACE2? If not, how conserved is the epitope in bats when compared to the target species of this antibody?

We thank the reviewer for their comment. We have moved this information from the supplementary methods section to the main revised manuscript (lines 185-209).  All details on the procedure as well as reagent information (reagent catalog number, manufacturer and dilution used) are clearly stated in the revised manuscript.  It is important to note that currently, commercially developed antibodies specific for bat ACE2 proteins are not available.  Secondly, although human and PB/ CVB ACE2 show a ~80% identity across the deduced protein sequences (Supplementary Table 1), key residues in the region 18-137 (which the anti-ACE2 antibody (Catalog # TA803844, Origene, Rockland, USA) binds to) are substantially different between the human and or PB or CVB ACE2 sequences (Supplementary Figure 1). These differences suggest poor binding of this antibody to bat ACE2 sequences and consequent low signal on the western blot. Also as raised by the reviewer in the next comment, we noticed  differences in the ACE2 transcript expression between PB and CVB ACE2 cells; these differences could be due to differential integration efficiency in the DF1 genome, insertion in transcriptionally silent sites or a combination of both. Together, differential ACE2 RNA expression, poor conservation of antibody binding site between human and PB or CVB ACE2 explain the low signal observed in the western blot. 

  • Authors explain in line 242 that ACE2 expression is evaluated via RT-qPCR but they do not include a figure summarizing this result. This information would be useful considering how weak ACE2 detection is via western-blotting.
  • We thank the reviewer for their interest in this correlation. We measured expression of ACE2 in both PB and CVB cell lines but did not show this data in the original manuscript because we believe that protein expression is more relevant to SARS-CoV-2 infection in these cell lines.  This data has now been included in the revised manuscript as supplemental material; compared to CVB which showed robust ACE2 transcript expression, we observed lower expression of PB ACE2 RNA (Supplementary Figure 2D). It is important to note that this could be either due to the lower insertion efficiency of the PB-ACE2 construct in this cell line, insertion into a transcriptionally inactive locus or a combination of both.  Western blot data however clearly shows a band of expected size in both PB and CVB ACE2 cells validating ACE2 protein expression in these cells.

  • I consider that the conclusion obtained from this study are correct according to the obtained results, however it seems contradictory with their own previous study (“SARS-CoV-2 utilization of ACE2 from different bat species allows for virus entry and replication in vitro”, PMID: 37542819).
  • This study demonstrated that ACE2 genes from CVB and to a lesser extent PB support infection and replication of SC2 variants. The prior study (PMID 37542819)  compared replication of WA1, delta and lambda but not Omicron variants of SC2 in the different bat ACE2 expressing cell lines.  Similar to our data, replication of SC2 variants was reduced in cells expressing bat ACE2 compared to the human ACE2/ TMPRSS2 pair.  
  •  

MINOR

  • Supplementary Figure 1, please indicate within the figure the three main regions with 100% shared identity that are mentioned in lines 225-226 using a red box or other visual element.

In the legend to Supplementary Figure 1, we clearly state that the green boxes represent regions of 100% identity between human , PB and CVB ACE2 sequences. For further visual clarity,  we have indicated regions on 100% identity with red boxes as recommended.

  • Lines 228-229: is the extent of the regions described correctly? For instance, first region seems to extend from position 266 to 277 according to the identity color code.
  • Please see response above.
  • Supplementary Figure 2 A and B are a composite of fluorescence acquired from two different channels in red (human TMPRSS2) and green (bat ACE2). Please indicate within each picture frame what are the colors represents to simplify the understanding of the pictures. Also, the scale bar value is too small.
  • We have revised the Figure 2A and 2B  to make it easier to comprehend.
  • A MOI = 1 is an abnormally high MOI to use for SARS-CoV-2 infection. Why authors used such a high MOI?
  • We used an MOI =1.0 for the following reasons.
    • The PB-ACE2 and CVB-ACE2 populations are not 100% pure as evident in IFA. Thus, it was important to potentially try and infect all cells that express the foreign bat ACE2 gene. Our qRT-PCR and western data also suggest low ACE2 expression in these cell lines. To overcome these limitations, we choose to use a higher MOI.
    • As referred to by the reviewer below, we also expected the affinity of bat ACE2 for SC2 RBD to be significantly lower than the human counterpart. Since this has not been determined for bat ACE2, a higher MOI would ensure some binding and infection.
    • All our assay end at 72 hour post infection and were performed in BSL3 settings for biosafety and biosecurity reasons. Also, except for qPCR, our IFA end points are expected to be less sensitive than PCR and thus we determined a high MOI infection would be most informative.
  • Lines 261-264, authors explain that differential infectivity between human and bat ACE2 cell lines can be explained by the use of different insertion systems, which is true. However, the affinity of bat ACE2 for SARS-CoV-2 RBD is significantly lower than human ACE (reported in PMID: 33335073). This is another parameter that should be taken into account, which in my opinion, has more influence than the transfection system.
  • We agree that the S protein: ACE2 affinity is a critical determinant of infectivity. However, considering that SC2 and variants were  grown in Vero E6 cells we believe that the viruses would be more adapted to growth in cell culture compared to clinical isolates. Thus, until the binding affinities of bat ACE2 genes and SC2 variant spike proteins are clearly defined, the primary explanation for differential integration efficiency and insertion between  transcriptionally inactive cs active sites remains the easiest explanation for differential infectivity and replication.
  • Include in Materials and Methods which primer sequences were used to evaluate ACE2 expression.
  • We stated those details in lines 183-185 and cite our previous study where the primer sequences were detailed. We have also included them in the revised version.
  • Lines 282-284, the text located here must be removed from the final version.

We have corrected this inadvertent omission.

Reviewer 2 Report

Comments and Suggestions for Authors

The working group of D. Kapczynskia has demonstrated in recent papers that human TMPRSS2 and ACE2 from different species render the chicken fibroblast cell line DF-1 susceptible to SARS-CoV-2 infection in vitro. Among others they found that ACE2 from seven different bat species supports entry of SARS-CoV-2, although less efficient compared to human ACE2. Furthermore, they showed that the Wuhan-like SARS-CoV-2 virus replicated to higher titers in DF-1 expressing bat ACE2s than Delta and Lambda variants.  The present manuscripts by Bakre et al. adds to these studies and shows that the ACE2 protein of two more bat species, common vampire bat (Desmodus rotundus) and pallid bat (Antrozous pallidus), supports SARS-CoV-2 infection of DF-1 cells. Further, the authors show that while Wuhan-like SARS-CoV-2 and Delta variant replicate efficiently in DF-1 expressing bat ACE2, Lambda and Omicron replicate only poorly in these cells.  The study is not new but complements previous studies well. A few points still need to be clarified:

Major:

Figure/Table S1-S3: The two bat proteins analyzed in this work are missing in the figures and sequence comparisons in S1 to S3. Please complete.

Fig. 2: Omicron generally grows poorly in the DF-1 cells regardless of which ACE2 protein is expressed. Is it ensured that Omicron is efficiently activated by human TMPRSS2? How does Omicron replicate in the cells when trypsin is added.

Fig. 3/4: Some of the cells are significantly less dense than the specified 75%. Why is that? Were the cells originally all equally confluent and were lost during staining?

Minor:

The abbreviation “SC2” for severe acute respiratory syndrome virus 2 is unusual. It would be better to use the common abbreviation “SARS-CoV-2”.  

Fig. 2: The green coloring of the Omicron graph varies.

Lines 282-284: The sentences probably originate from the template and must be deleted.

Author Response

Reviewer 2.

The working group of D. Kapczynskia has demonstrated in recent papers that human TMPRSS2 and ACE2 from different species render the chicken fibroblast cell line DF-1 susceptible to SARS-CoV-2 infection in vitro. Among others they found that ACE2 from seven different bat species supports entry of SARS-CoV-2, although less efficient compared to human ACE2. Furthermore, they showed that the Wuhan-like SARS-CoV-2 virus replicated to higher titers in DF-1 expressing bat ACE2s than Delta and Lambda variants.  The present manuscripts by Bakre et al. adds to these studies and shows that the ACE2 protein of two more bat species, common vampire bat (Desmodus rotundus) and pallid bat (Antrozous pallidus), supports SARS-CoV-2 infection of DF-1 cells. Further, the authors show that while Wuhan-like SARS-CoV-2 and Delta variant replicate efficiently in DF-1 expressing bat ACE2, Lambda and Omicron replicate only poorly in these cells.  The study is not new but complements previous studies well. A few points still need to be clarified:

 We thank the reviewers for their time and effort in improving the manuscript. Our responses to the comments are listed in blue below.

Major:

  • Figure/Table S1-S3: The two bat proteins analyzed in this work are missing in the figures and

sequence comparisons in S1 to S3. Please complete.

The two proteins of focus here (PB and CVB) ACE2 are prominently highlighted in Supplementary Table 1 and  in Supplementary Figure 1 and 2 as well.

  • 2: Omicron generally grows poorly in the DF-1 cells regardless of which ACE2 protein is expressed. Is it ensured that Omicron is efficiently activated by human TMPRSS2? How does Omicron replicate in the cells when trypsin is added.

Both the PB-ACE2 and CVB-ACE2 cell lines express human TMPRSS2 constitutively so we expect efficient activation. Data from Figure 2C, shows that Hs-ACE2 and Hs-TMPRSS2 are sufficient for robust Omicron replication. The PB-CV2 and CVB-ACE2 cells also express human TMPRSS2. Thus, data suggest that lack of Omicron replication in PB-ACE2 and CVB-ACE2 is primarily driven by poor binding to the respective ACE2 receptors. Thus, we did not test Omicron replication with trypsin since it was out of scope for the study.

  • 3/4: Some of the cells are significantly less dense than the specified 75%. Why is that? The images in Figures 3 and Figure 4 are captured from chamber slides in containment.
  • Cell numbers were deliberately kept low to ensure optimal signals and prevent overcrowding of cells.

Minor:

The abbreviation “SC2” for severe acute respiratory syndrome virus 2 is unusual. It would be better to use the common abbreviation “SARS-CoV-2”.  

We continued with our abbreviation of SC2 to keep consistency with our previous publications.

Fig. 2: The green coloring of the Omicron graph varies.

We thank the reviewers for their time and effort in improving the manuscript. We have made sure all graphs are now consistent.

Lines 282-284: The sentences probably originate from the template and must be deleted.

Thank you. We have removed the indicated statements in the revised version of the manuscript.

Round 2

Reviewer 1 Report

Comments and Suggestions for Authors

Authors have properly addressed my concerns. However, there are some minor errors that must be corrected before the final version of the manuscript is accepted.

  • In the PDF file, captions from figure 1 and 2 are not properly located below each figure.
  • Similarly, letters (A, B and C) associated with each plots in Figure 2 are out of position, they should be located upper right of each plot instead of bottom right.
  • Author's response to one of my comments was: "We performed control experiments with just DF-1 cells that lack ACE2 or TMPRSS2 and observed no replication of SC2 variants in wild type DF-1 cells as shown previously." I understand that these infection controls in DF-1 cells were performed in a previous study. If this is the case, please cite the respective study at the beginning of section 3.2. This has to be explicitely mentioned, otherwise readers might think that these controls were never performed.
  • Figure 2C, the color code for variants shown in this graph is not consistent with the color code used previously in Figure 2A and 2B. This might be confusing for readers.
  • There is no reference to the new Supplementary Figure 2D across the manuscript. Please, refer to this figure when mentioning analysis of ACE2 expression via qPCR.

Author Response

Authors have properly addressed my concerns. However, there are some minor errors that must be corrected before the final version of the manuscript is accepted.

We thank the reviewer for their time and help in improving the quality of this manuscript. Listed below are our responses to their comments.

  • In the PDF file, captions from figure 1 and 2 are not properly located below each figure.

We have fixed the formatting issue in the revision.

  • Similarly, letters (A, B and C) associated with each plots in Figure 2 are out of position, they should be located upper right of each plot instead of bottom right.

We have arranged the sub-panel labels in the top left corner of each panel in alignment with other published manuscripts in Viruses. We have also increased the font size of the magnification scale and metrics for ease of readership.

  • Author's response to one of my comments was: "We performed control experiments with just DF-1 cells that lack ACE2 or TMPRSS2 and observed no replication of SC2 variants in wild type DF-1 cells as shown previously." I understand that these infection controls in DF-1 cells were performed in a previous study. If this is the case, please cite the respective study at the beginning of section 3.2. This has to be explicitly mentioned, otherwise readers might think that these controls were never performed.

Thank you for the clarification. We have now added this explicit statement to the beginning of section 3.2 (Line 269).

Figure 2C, the color code for variants shown in this graph is not consistent with the color code used previously in Figure 2A and 2B. This might be confusing for readers.

Thank you for pointing this out. We have now changed the color schema to be very clearly distinguishable and consistent across all panels.

  • There is no reference to the new Supplementary Figure 2D across the manuscript. Please, refer to this figure when mentioning analysis of ACE2 expression via qPCR.

Thank you for pointing this out. Reference to Figure 2D has now been included in line 273 of the revised manuscript.